# Cytoprotective Effects of Natural Highly Bio-Available Vegetable Derivatives on Human-Derived Retinal Cells

**DOI:** 10.3390/nu12030879

**Published:** 2020-03-24

**Authors:** Ingrid Munia, Laurent Gafray, Marie-Agnès Bringer, Pablo Goldschmidt, Lil Proukhnitzky, Nathalie Jacquemot, Christine Cercy, Khaoula Ramchani Ben Otman, Marie Hélène Errera, Isabelle Ranchon-Cole

**Affiliations:** 1UMR1107 NeuroDol, Université Clermont-Auvergne, TSA 50400, 28 Place Henri Dunant, 63000 Clermont-Ferrand, France; il.munia@gmail.com (I.M.); laurent.gafray@laposte.net (L.G.); Nathalie.JACQUEMOT@uca.fr (N.J.); Christine.CERCY@uca.fr (C.C.); rbo.khaoula@live.fr (K.R.B.O.); 2Centre des Sciences du Goût et de l’Alimentation, AgroSup Dijon, CNRS, INRAE, University of Bourgogne Franche-Comté, Eye and Nutrition Research Group, F-21000 Dijon, France; marie-agnes.bringer@inra.fr (M.-A.B.); lil.proukhnitzky@gmail.com (L.P.); 3Centre Hospitalier National d’Ophtalmologie des Quinze-Vingts, 75012 Paris, France; pablogol@aol.com (P.G.); errera.mhelene@gmail.com (M.H.E.)

**Keywords:** retina, damage, protection curcumin, resveratrol, lutein, antioxidant, apoptosis, autophagy

## Abstract

Retinal pigment epithelial cells are crucial for retina maintenance, making their cytoprotection an excellent way to prevent or slow down retinal degeneration. In addition, oxidative stress, inflammation, apoptosis, neovascularization, and/or autophagy are key pathways involved in degenerative mechanisms. Therefore, here we studied the effects of curcumin, lutein, and/or resveratrol on human retinal pigment epithelial cells (ARPE-19). Cells were incubated with individual or combined agent(s) before induction of (a) H_2_O_2_-induced oxidative stress, (b) staurosporin-induced apoptosis, (c) CoCl_2_-induced hypoxia, or (d) a LED-autophagy perturbator. Metabolic activity, cellular survival, caspase 3/7 activity (casp3/7), cell morphology, VEGF levels, and autophagy process were assessed. H_2_O_2_ provoked a reduction in cell survival, whereas curcumin reduced metabolic activity which was not associated with cell death. Cell death induced by H_2_O_2_ was significantly reduced after pre-treatment with curcumin and lutein, but not resveratrol. Staurosporin increased caspase-3/7 activity (689%) and decreased cell survival by 32%. Curcumin or lutein protected cells from death induced by staurosporin. Curcumin, lutein, and resveratrol were ineffective on the increase of caspase 3/7 induced by staurosporin. Pre-treatment with curcumin or lutein prevented LED-induced blockage of autophagy flux. Basal-VEGF release was significantly reduced by lutein. Therefore, lutein and curcumin showed beneficial protective effects on human-derived retinal cells against several insults.

## 1. Introduction

Age-related macular degeneration (AMD) is a leading cause of vision loss in the population over 65 [1]. Its pathogenesis involves a complex interaction of metabolic, functional, genetic, and environmental factors. The mechanisms associated with AMD combine oxidation, inflammation, hypoxia, and neovascularization of retinal tissues [1]. Abnormalities affect functional interrelated tissues, i.e., photoreceptors, retinal pigment epithelium (RPE), Bruch’s membrane, and choriocapillaries. However, the impairment of RPE cell function is an early and crucial event in the molecular pathway leading to AMD changes, suggesting that enhancing RPE cell resistance to stress is of significant interest. Therefore, in the present study, we investigated the ability of bio-available vegetable derivatives to protect human retinal pigment epithelial cells (ARPE-19) against several stressors involved in AMD pathogenesis [2]. This study is based on data obtained from prospective studies (Age-Related Eye Disease Studies, AREDS) [3,4] bringing to light the protective effect of nutritional supplements on the progression of ocular diseases. Indeed, AREDS1 2001 showed that daily oral supplementation with antioxidants (β-carotene), vitamins (C and E), and minerals (copper) reduced the risk of developing AMD by 25% at 5 years [3]. In the AREDS2 trial carried out in 2013, β-carotene was substituted by lutein and zeaxanthin, since β-carotene increased lung cancer incidence among smokers [4]

The phytochemicals chosen were lutein, curcumin, and resveratrol. Lutein, as zeaxanthin, exists broadly in the retina [5]. They cannot be synthesized within human cells and should be provided through a diet of leafy green vegetables, some fruits, and egg yolks [6]. Its intake has been suggested in recent nutritional recommendations [7]. They accumulate in the inner retinal layer of the macula [8]. They participate in the absorption of harmful blue light and contribute to the limiting of oxidative stress in RPE cells [8,9,10]. Moreover, lutein and zeaxanthin exert neuroprotective effects in vision-threatening diseases, such as innate retinal inflammation, diabetic retinopathy, and participation in the development and/or maintenance of the RPE, which could thereby reduce susceptibility to macular degeneration [11]. Recently, lutein has been shown to modulate autophagy [12], an intracellular degradation system that delivers cytoplasmic constituents to the lysosome [13]. Autophagy mediates numerous cytoprotective effects [14]. Dysregulated autophagy has been associated with increased susceptibility to oxidative stress in the RPE and is considered as a contributing factor in AMD [15]. 

Resveratrol (trans-3, 5, 40-trihydroxystilbene) is a natural polyphenolic phytoalexin found in grape skins [16]. It has anti-oxidative [17,18,19] and anti-inflammatory [20,21] effects and can modulate autophagy [22]. Protective effects have been reported for several ocular diseases, such as light/oxygen-induced retinopathy [23,24], endotoxin-induced uveitis [20], and diabetic retinopathy [24,25]. 

Addition of compounds, such as curcumin, a natural phenolic compound extracted from ground rhizomes of *Curcuma longa L*. (*Curcuma domestica* Valeton), could ameliorate the efficiency of nutritional supplements. Curcumin has antioxidant, anti-microbial, anti-inflammatory, anti-proliferative, and pro-apoptotic effects [26,27], as well as multiple direct and indirect targets, including enzymes, apoptosis-related proteins, adhesion molecules, inflammatory cytokines, growth factors, protein kinases, and transcription factors [28]. Beneficial effects of curcumin has been reported in several pathophysiological contexts, such as Alzheimer Disease [29], multiple sclerosis [30], Parkinson’s disease [31], epilepsy [32], cerebral injury [33], and age-associated neurodegeneration [34]. In addition, benefits of curcumin for ocular diseases have been pointed out [35,36]. In cultured human RPE cells, curcumin was able to inhibit cell proliferation by triggering caspase 3/7-dependent (but caspase 8-independent) cell death and necrosis [37]. In addition, curcumin has been shown to modulate autophagy [38].

Therefore, in the present study, we investigated the ability of lutein, resveratrol, and curcumin to protect human retinal pigment epithelial cells (ARPE) against environmental stress. For that purpose, ARPE-19 cells were pre-treated with lutein, resveratrol, or curcumin, individually or combined. Then, they were exposed to oxidative stress, apoptosis, hypoxia, or blue light, and cell alteration was evaluated by analyzing mitochondrial activity, cell survival, caspase 3/7 activity, VEGF level, and autophagy activity. 

## 2. Materials and Methods 

**Cell line and culture conditions.** The human retinal pigment epithelial cells ARPE-19 were obtained from the American Type Culture Collection (ATCC, Manassas, VA, USA). ARPE-19 were maintained in a 1:1 mixture, Dulbecco’s Modified Eagle’s Medium/Nutrient Mixture F-12 Ham (DMEMF-12, Gibco, Life Technologies, Carlsbad, USA), supplemented with 10% of fetal bovine serum (Gibco, Life Technologies, Carlsbad, CA, USA) and 1% of penicillin-streptomycin (Gibco, Life Technologies, Carlsbad, USA) in an atmosphere containing 5% CO_2_ at 37 °C. For experiments, cells were seeded in 96-well or 24-well plates at a concentration of 2.10^5^ cells/mL, and maintained in an atmosphere containing 5% CO_2_ at 37 °C for 48 h before their use.

**Experimental Paradigm:** For experiments, cells were seeded in 96-well or 24-well plates at a concentration of 2.10^5^ cells/mL and maintained in an atmosphere containing 5% CO_2_ at 37  °C for 48 h. Then, they were pretreated with or without the natural agent(s) for 24 hours before being subjected to H_2_O_2_ (oxidative stress), staurosporin (apoptosis), CoCl_2_ (hypoxia), and LED exposure (autophagy alteration). The susceptibility of ARPE19 cells in these conditions was evaluated by measuring metabolic activity (MTT), cell viability (Neutral red or Apotox Kit), caspase 3/7 activity, VEGF secretion (ELISA), protein expression (immunoblot), and/or structure integrity (immunocytochemistry). 

**Pre-treatments.** Lutein FloraGlo® 10% is a water-soluble formula, where resveratrol 95% and highly bioavailable curcumin 98% were provided by Densmore, Monaco. Resveratrol and curcumin were solubilized in DMSO 100% (Sigma-Aldrich St. Louis, MO, USA) and lutein in water before dilution in culture medium. Fresh culture medium containing lutein, resveratrol, or curcumin was added for a 24 h period. The maximal final concentration of DMSO present in cultures was 0.1%.

**Calibration of oxidative Stress induced by H_2_O_2._** ARPE-19 cells were treated with H_2_O_2_ µM (30% solution Sigma, France) at a final concentration of 0 to 1200 µM in serum-free media. Cells were then incubated at 37 °C for 2 hours before metabolic activity assessment. Results were plotted as relative absorbance as a function of H_2_O_2_ concentrations. Each curve was fitted with Origin 6 (Microsoft) using the Boltzmann Growth/Sigmoidal function to calculate IC5_0_:
$y=A_{2}=(A_{1}-A_{2})/(1+e^{\left(x-x_{0})/dx\right)})$. *A_1_*, *A_2_*, *x_0_*, and *dx* are four calculated parameters, and x is the concentration of H_2_O_2_.

**H_2_O_2_-induced oxidative stress on pretreated cells.** Pre-treated or untreated ARPE-19 cells were subjected to H_2_O_2_ 600 µM in serum-free media for 2 hours. Cell metabolic activities were studied using the MTT assay.

**Assessment of cell metabolic activity by MTT assay**. ARPE-19 cells were washed with PBS (Phosphate Buffer Solution PH: 7.4) and incubated with fresh medium containing 0.5 mg/mL of 3-(4,5-dimethylthiazol-2-yl)-2,5-diphenyl tetrazolium bromide (MTT, Sigma-Aldrich, St Louis, MO) for 2 h. MTT was removed and cells were rinsed with PBS before dissolving purple formazan crystals in 150 µL of DMSO 100 %. The absorbance of the supernatants was read at 570 nm. 

**CoCl_2_-induced hypoxia**. Pre-treated or untreated ARPE-19 cells were subjected to CoCl_2_ at 200 µM for 4 h. The supernatant was collected in order to perform VEGF quantification by ELISA. Viability and caspase 3/7 enzymatic activity (Apotox) was assessed on cells.

**Quantitative estimation of viable cells by Neutral Red uptake**. ARPE-19 cells were incubated for 2 h 30 min in an atmosphere containing 5% CO_2_ 37 °C with neutral red (Sigma) at 0.04 mg/mL in cell culture medium. Viable cells incorporate and bind neutral red in the lysosomes. Cells were then washed with PBS, and 100 µL of a de-staining solution (1% acetic acid, 50 % ethanol in H_2_O) were added. Plates were gently shacked for 10 minutes before reading absorbance at 540 nm.

**Staurosporin-induced apoptosis** Pretreated or untreated ARPE19 cells were incubated with staurosporin at a concentration of 1 µM in complete cell culture media for 18 h. Cell viability and caspase 3/7 activity were assessed by using ApoTox-Glo™ Triplex Assay (Promega Corporation) according to the manufacturer’s instructions. Viable cells were quantified using a cell-permeant fluorogenic peptide substrate (GF-AFC Substrate) that penetrates into intact cells, where it is cleaved to generate a fluorescent signal, which is proportional to the number of living cells. This live-cell protease activity marker becomes inactive upon loss of membrane integrity. Then, the luminogenic DEVD-peptide substrate for caspase-3/7, combined with Ultra-Glo™ recombinant thermostable luciferase was added. Substrate cleavage by caspase-3/7 releases luciferin, the substrate for luciferase, which generates light that correlates to caspase-3/7 activation. Fluorescence and luminescence were measured using Ascent Fluoroscan (Thermo Labsystem, France). Results are presented as relative fluorescent or luminescence signals. 

**LED-induced Autophagy blockage** ARPE-19 cells were seeded as 1.8 mL per well of a suspension at 1.10^5^ cells/mL in 24-well plates. Two days later, cells were untreated or treated with lutein at 3.5 µM or curcumin at 3.4 µM for 24 hours. Fresh cell culture media was replaced with 1.8 mL of media without any treatment. Cells were washed before adding fresh cell culture media. Then, they were kept in the dark or exposed to blue LED for 4 hours. Twenty-four hours later, autophagic flux was evaluated. 

**Autophagy analysis by immunoblot**: Cells were either treated or untreated with bafilomycin A1 (LC Laboratories), an inhibitor of vacuolar-type H^+^ ATPases, at 150 nM for 2 h prior to cell lysis and protein extraction. Whole-cell protein extracts were prepared by using RIPA lysis and extraction buffer (Thermo Fisher Scientific). Briefly, proteins were separated on 4%–15% Mini-PROTEAN® TGXTM precast protein gels (Bio-Rad), transferred to nitrocellulose membrane, blocked for 2  h in phosphate-buffered saline (PBS) solution containing 5% nonfat dry milk, probed overnight at 4 °C with primary antibodies directed against LC3B (Sigma L7543, 1/1000) or Actin (Sigma A5060, 1/1000), and for 2 h at room temperature with secondary HRP-coupled antibodies (DAKO P0448). After membrane revelation using the ECL detection kit (Bio-Rad), quantification was performed with the Image Lab software (Bio-Rad).

**Quantification of Vascular endothelial growth factor** (VEGF) by enzyme-linked immunosorbent assay (ELISA). The supernatants of lutein, resveratrol or curcumin pre-treated or untreated cells were collected and concentration of VEGF was determined using the Human VEGF ELISA Kit (KHG0111; Invitrogen Novex®) according to the manufacturer’s instructions. Briefly, media were centrifuged at 1400 rpm at 4 °C, and 50 µL of supernatant was added to 50 µL of a standard diluent into a 96-well plate pre-coated with a polyclonal anti-VEGF antibody. Hu-VEGF biotin conjugate, followed by a Streptavidin-HRP solution, was added, and after 30 minutes at room temperature, the wells were washed before adding the chromogen. After dark incubation for 30 min at room temperature, the stop solution was added, and absorbance was read at 450 nm. 

**Immunocytochemistry staining**. Cells were seeded at 1.8 mL per well into a 24-well plate containing glass cover plates for 48 h and then treated with curcumin, lutein, or curcumin + lutein for 24 hours. Nuclei were labeled with DAPI (Sigma-Aldrich) at a dilution of 1/500, and F-actin was labeled using Phalloidin Alexa Fluor 568 (Invitrogen). Images were obtained using a confocal microscope ((Leica SP5; Leica Corp., Wetzlar, HE, Germany) and analyzed with Image J software. Results indicate means ± SEM. Analysis of variance (ANOVA) was performed using *Statistica* (StatSoft, Inc). If ANOVA was significant, multiple comparisons were conducted to assess pairs of mean values, and differences between groups were studied using the post hoc Turkey test (significance set for *p* = 0.05). For autophagic data, statistical analyses were performed with GraphPad Prism software (GraphPad Software Inc., San Diego, CA, USA). The non-parametric Mann–Whitney test was used to compare results between cells with and without bafilomycin A1. The non-parametric Kruskal–Wallis test was used for multiple comparisons between conditions (untreated cells, and cells treated with lutein and curcumin). *P* values ≤ 0.05 were considered as statistically significant. There is one symbol for *p* < 0.05, two symbols for *p* < 0.01, three symbols for *p* < 0.001, and four symbols for *p* < 0.0001.

## 3. Results

### 3.1. Oxidative Stress Induced by H_2_O_2_

#### 3.1.1. Calibration of H_2_O_2_ Effects

Depending on the experiment, plates had to be seeded with 100 or 300 µL of cell suspension, resulting in a different number of cells to start with. Therefore, the first step consisted of evaluating the sensitivity of cells to H_2_O_2_ in these different conditions (Figure 1). The effect of H_2_O_2_ was quantified by MTT measuring the metabolic activity and neutral red for cell survival [39]. The concentration of H_2_O_2_ leading to metabolic activity reduction of 50% was 560 ± 16 µM for 300 µL suspensions (6.10^4^ cells) and 610 ± 20 µM for 100 µL (2.10^4^ cells) (Boltzmann Growth/Sigmoidal function) (Figure 1). Therefore, in the following experiment we used 600 µM H_2_O_2_ to induce oxidative stress. 

#### 3.1.2. Effects of Pretreatment on Cell Susceptibility to Oxidative Stress Induced by H_2_O_2_


Since the H_2_O_2_ effect was quantified by measuring metabolic activity by MTT, we first analyzed the effects of curcumin, resveratrol, and lutein on the basal metabolic activity of ARPE-19 cells. Curcumin significantly (*p* < 0.05) decreased basal metabolic activity at doses from 0.68 to 27.17 µM (Figure 2a), but lutein (Figure 2c) and resveratrol (Figure 2e) had no effect. Because curcumin affected metabolic activity, we analyzed the morphology of ARPE-19 cells by confocal microscopy using labelling with DAPI/Phalloidin. Labelling did not reveal cell structure disorganization or necrosis (Figure 3) since membrane integrity was preserved.

Then, we analyzed the effect of H_2_O_2_. At 600 µM, H_2_O_2_ induced a significant (*p* < 0.01) reduction of metabolic activity (51 to 25%, *p* < 0.0005) (Figure 2a–c) in cells without pretreatment. Pretreatment with curcumin up to 0.34 µM did not significantly modify the reduction of cell metabolic activity induced by H_2_O_2_ (Figure 2b). However, H_2_O_2_ had a significantly (*p* < 0.05) less effect on metabolic activity of cells pretreated with curcumin at 0.68 up to 27.17 µM (Figure 2b) or with lutein at 1.8 µM and 3.5 µM (Figure 2d). Resveratrol pretreatment did not prevent the reduction of cell metabolic activity induced by H_2_O_2_ (Figure 2f).

These results showed that curcumin and lutein reduced the effect of oxidative stress induced by H_2_O_2_. In addition, we can observe that only a concentration of the agent leading to a reduction of basal activity was significantly able to significantly reduce the effect of H_2_O_2_, suggesting a correlation between a reduced basal metabolic activity and a resistance to oxidative stress. 

#### 3.1.3. Effect of Combination of Natural Agents on Cell Susceptibility to Oxidative Stress Induced by H_2_O_2_

Further, the susceptibility to H_2_O_2_ was analyzed in cells pretreated with a combination of these natural agents (curcumin + lutein or curcumin + lutein + resveratrol) (Figure 4) in order to evaluate a synergistic or antagonist effect. The concentrations used were based on the previous experiments. Curcumin and lutein concentrations corresponded to the one affecting cell sensitivity to H_2_O_2_ (curcumin at 0.34 or 3.4 µM and lutein at 3.5 µM) and for resveratrol, we used the highest concentration (43.6 µM). In order to calculate the reduction of metabolic activity induced by H_2_O_2_ we subtracted the metabolic activity of cells with H_2_O_2_ to the one of cells without H_2_O_2_ for each treatment (Figure 4). The results showed that pretreatment with curcumin + lutein ± resveratrol significantly limited (*p* < 0.01) the reduction of metabolic activity induced by H_2_O_2_. However, there was no synergistic or antagonist effect, since the results of the combined treatment were similar to that of the single treatment (Figure 2).

### 3.2. Apoptosis Induced by Staurosporin

#### 3.2.1. Calibration of Staurosporin Effects 

Cell metabolic activity and cell viability were evaluated on staurosporin-treated ARPE-19 cells by an MTT or neutral red assay, respectively (Figure 5). As expected, staurosporin induced a dose-dependent decrease in cell metabolic activity associated with a reduction of cell viability. IC50 and IC80 were determined from dose-response data. IC50 was 3.6 ± 9 µM for cell metabolic activity and 6.1 ± 0.7 µM for cell viability. IC80 was 0.7 ± 0.2 µM for metabolic activity and 1.1 ± 0.1 µM for cell viability. In further experiments, we will choose to use the dose of 1 µM of staurosporin to limit drastic effects. 

#### 3.2.2. Effects of Pretreatment on Cell Susceptibility to Apoptosis Iduced by Staurosporin 

First, we evaluated the effect of pretreatment on basal cell viability and caspase 3/7 activity. Curcumin, lutein, or resveratrol had no significant impact on ARPE-19 cell survival (Figure 6a). Basal caspase 3/7 activity was increased by curcumin or lutein, but values were not significantly different from untreated cells (Figure 6b). However, caspase activity was significantly enhanced (*p* < 0.03) by resveratrol (405 ± 96 % at 0.2 µM; 479 ± 79% at 1.1 µM and 420 ± 58 % at 5.5 µM). 

Second, we studied the effect of pretreatment on ARPE19 susceptibility to staurosporin. Staurosporin significantly reduced cell survival (Figure 6c; 1µM staurosporin-concentration 0) and induced a significant activation of caspase 3/7 (Figure 6d, 1µM staurosporin-concentration 0) Curcumin, lutein, or resveratrol pre-treatment did not reduce activation of caspase 3/7 induced by staurosporin. However, we observed that staurosporin did not significantly affect survival of cell pretreated with curcumin at doses of 0.34 and 3.4 µM, nor with lutein at all tested doses, but induced a significant decrease of survival in cells pretreated with resveratrol. 

These results suggest that curcumin and lutein, but not resveratrol protects human retinal cells from apoptosis induced by staurosporin. In addition, this protection does not involve the caspase 3/7 pathway. It should be noted that high concentrations of curcumin (27 µM) could have toxic effects and lead to a large caspase 3/7 activation (Figure 6c) associated with cell death (Figure 6d).

#### 3.2.3. Effects of Combination of Natural Agents on Cell Susceptibility to Apoptosis Induced by Staurosporin

Further, the effect of a combination of natural agents (curcumin + lutein or curcumin + lutein + resveratrol) on caspase 3/7 activity and apoptosis was tested (Figure 7). Pretreatment with curcumin 0.34 µM + lutein 3.5 µM, or with curcumin 3.4 µM + lutein 3.5 µM had no effect on cell survival (Figure 7a), but it significantly increased caspase 3/7 activity by 396 ± 72 % (*p* = 0.006) and 400 ± 59 % (*p* = 0.01), respectively (Figure 7b). The addition of resveratrol reduced cell survival to 81 ± 3 % (*p* = 0.04) for curcumin 0.34 µM + lutein 3.5 µM + resveratrol and to 73 ± 6 % (*p* = 0.003) for curcumin 3.44 µM + lutein 3.5 µM + resveratrol, and increased caspase-3/7 activity to 539 ± 62 % and 525 ± 60 %, respectively (Figure 7b). 

Thereafter, we evaluated the effect of combination of natural agents on caspase 3/7 activity and cell death induced by staurosporin in ARPE-19 cells. Staurosporin had no significant effect on survival in cells pretreated with curcumin + lutein. However, these pretreatments did not reduce caspase activation induced by staurosporin (Figure 7b). Therefore, pretreatment with a combination of curcumin + lutein reduced cell death independently from caspase 3/7 activity. There were no synergistic or antagonist effects between lutein and curcumin (compared to Figure 6). 

### 3.3. Effects of Pretreatment on the Alteration of Autophagy Flux Induced by Blue LED

Autophagy is a dynamic process. To accurately measure autophagy, we performed an autophagy flux assay by quantifying accumulation of LC3-II protein in cells treated or untreated with bafilomycin A1, a compound that blocks the autophagy flux. Briefly, accumulation of LC3-II in cells with bafilomycin A1, compared to cells without bafilomycin A1 indicates the production of new autophagic vacuoles [40]. In ARPE-19 cells not exposed to blue LED, we observed that lutein induced a slight but non-significant increase in the amount of LC3-II compared to untreated cells, both in conditions without and with bafilomycin A1 (Figure 8A,B). No significant effect of curcumin on autophagy was observed in these conditions (Figure 8A,B). 

Then, we analyzed the effect of blue LED exposure on autophagy. We observed a 2.7-fold higher accumulation of LC3-II in ARPE-19 cells exposed to blue LED comparatively to non-exposed cells (*p* < 0.05), in a bafilomycin A1-free condition (Figure 8C,D). This result suggested that blue LED could induce autophagy. However, comparison of the LC3-II amounts in cells with and without bafilomycin A1 revealed that the autophagy flux was altered by cell exposure to blue LED (Figure 8C,D). Indeed, whereas a significant increase in the amount of LC3-II was observed in ARPE-19 cells not exposed to blue LED with bafilomycin A1 in comparison to those without bafilomycin A1, the amount of LC3-II remained similar with and without bafilomycin A1 in cells exposed to blue LED. 

The pre-treatment of cells with lutein, but not with curcumin, before blue LED exposure allowed for the maintenance of autophagy flux, as shown by the significant increase in the amount of LC3-II in cells exposed to blue LED with bafilomycin A1 in comparison to those without bafilomycin A1 (Figure 8C,D).

Altogether, these results show that blue LED exposure induces a blockade of the autophagy flux, and suggests that lutein could prevent this alteration.

### 3.4. Effects of Pretreatment on VEGF Secretion 

#### 3.4.1. Effects of Pretreatment on VEGF Secretion in Basal Conditions

ARPE-19 cells were pretreated with curcumin, lutein, resveratrol, or a mixture of 2 or 3 of these natural agents for 24 h before collecting cell supernatants for VEGF quantification (Figure 9). When an ANOVA statistical analysis was performed, no significant difference was observed between groups. This is probably because of inter-test variations. However, when we compared the effect of each molecule individually, we observed that cell treatment with lutein significantly (*p* = 0.0035) reduced the basal level of VEGF secreted in culture medium, and that cell treatment with resveratrol at 3.4 µM significantly (*p* = 0.002) increased it. No significant effect on VEGF level was observed for cells treated with curcumin at 3.5 µM. It should be noted that the decrease in VEGF amount observed in cells treated with resveratrol at the highest concentration is related to a reduced cell number. The combination of curcumin 3.4 µM + lutein 3.5 µM significantly (*p* = 0.04) decreased VEGF secretion. 

#### 3.4.2. Effects of Combined Pretreatment on VEGF Secretion in Hypoxia-Induced by CoCl_2_

Hypoxia was induced by incubating ARPE19 cells with 200 µM of CoCl_2_ for 4 hours. CoCl_2_ had no significant effect on cell viability or caspase 3/7 activity (Figure 10b,c) but induced a significant increase in VEGF release (Figure 10a) in untreated cells. Pretreatment significantly reduced basal VEGF release and increased caspase activity without affecting cell viability. CoCl_2_ did not increase VEGF release in pretreated cells. In the above, curcumin + lutein significantly reduced the release of VEGF induced by CoCl_2_.

## 4. Discussion

Inflammation, oxidative stress, and angiogenesis has been shown to highly contribute to AMD development or progression. More recently, autophagy has also been suggested as a contributing factor in AMD by playing a role in the regulation of RPE cell death [15]. Although no pharmaceutical medicines are available to treat AMD, natural agents with anti-inflammatory, anti-oxidant, and anti-angiogenic properties, such as omega-3 fatty acids, vitamins, and carotenoids have been reported to exert a protective effect against AMD [3,4]. As a consequence, numerous natural agents have been recommended as nutritional supplements in the prevention of ocular diseases [41]. However, only a few studies exist on the impact of these compounds at the cellular and molecular levels and their mode of action to protect retinal cells. 

In the present study, we analyzed the ability of nutritional supplements to protect human retinal pigment epithelial cells (ARPE-19 cells) against several harmful stressors (oxidative stress, induced cell death, hypoxia, and autophagy alteration induced by blue LED). The chosen natural agents were lutein, curcumin, and resveratrol. Lutein was chosen because its intake was suggested in nutritional recommendations [7], resveratrol because it has anti-oxidative [17,18,19] and anti-inflammatory effects [20,21] and can modulate autophagy [22], and curcumin because its beneficial effects had been reported in several neurodegenerative diseases [29,30,31,32,33,34] and its therapeutic potential in ophthalmology has been reported [35]. The doses of natural agents used in this study were selected according to their oral bioavailability and the conceivable amount in the blood circulation after oral administration. Curcumin has been shown to have low bioavailability [42,43]. FloraGlo formulation of lutein enhances bioavailability [44]. Resveratrol is well-absorbed, but its bioavailability is low [45]. Cells were treated at concentrations ranging from 0 to 27.17 µM for curcumin, 0 to 70 µM for lutein [46], and 0 to 43.9 µM for resveratrol [47]. 

We first evaluated the effect of lutein, curcumin, and resveratrol in basal conditions (no stressors). It had been shown in several studies that lutein and resveratrol reduced cell proliferation [48,49,50,51]. However, in our study, cells were treated when they reached confluency, and so the proliferation was very limited. As previously described, no deleterious effects of lutein [52] and resveratrol [48] were observed. For curcumin, we noticed a decrease in metabolic activity. This decrease could have resulted from a reduced number of cells, since curcumin had been shown to decrease cell proliferation at 50–70 µM [53] or a slowdown of metabolic activity per cell. The second hypothesis was confirmed, since curcumin did not affect cell survival. Consequently, curcumin reduced basal metabolic activity without modifying the cell number. This reduction in metabolic activity can contribute to the protective effects of curcumin. Indeed, it has been suggested that reduction in energy usage induced by hypothermia would provide protection during ischemia [54]. In addition, the results obtained here highlight the fact that metabolic activity and cell number do not always directly correlate, and so a metabolic activity assay, such as MTT, has to be used with caution for cell viability.

Further, we showed that lutein, resveratrol, and curcumin agents increased basal caspase 3/7 activity. Such activation has been previously reported in lung cancer-derived cell lines [55]. Interestingly, recent studies demonstrated that caspases can exert both apoptotic and non-apoptotic activities, and their activation does not exclusively lead to cell death [56,57,58]. Indeed, caspases are involved in regeneration, wound healing, proliferation, and in cell motility [59,60]. Such a phenotype could explain why we observed caspase activation without cell death. 

VEGFs are proteins secreted by a variety of cells, primarily acting on endothelial structures to initiate and promote angiogenesis during development and in pathological conditions, such as AMD and diabetic retinopathy [61,62,63]. Anti-VEGF therapies are presently one of the main focuses for the therapeutic management of AMD [64,65,66]. Among the natural agents tested, lutein (3.5 µM) was the only one that reduced the basal amount of VEGF secreted by ARPE-19 cells. In addition, we showed that in our conditions, neither lutein nor curcumin were able to induce a significant activation of autophagy under normal conditions. Autophagy is crucial for homeostasis maintenance in RPE cells, particularly by its role in the degradation of damaged proteins and organelles in the regulation of the inflammatory response and in the fight against oxidative stress [15,67]. 

Thereafter, we evaluated the susceptibility of cells pretreated with lutein, curcumin, or resveratrol to stress conditions. Curcumin, lutein, but not resveratrol reduced oxidative stress induced by H_2_O_2_ and apoptosis induced by staurosporin. The dose of curcumin that allows for protection of ARPE-19 cells from death induced by staurosporin was much lower than those used in primary rat hippocampal neuron cultures [68]. Nevertheless, neither curcumin nor lutein had effects on staurosporin-induced caspase 3/7 activity. These results confirm that caspase activation might not be directly associated with induction of apoptotic cell death. In the present study, we showed that exposure to blue LED induces accumulation of LC3-II in ARPE-19 cells. This result is in line with data published by others reporting accumulation of LC3-II in retinal cells in response to blue LED exposure [69,70,71]. However, by using an autophagy flux assay, we were able to show that the autophagy flux was altered in ARPE-19 cells exposed to blue LED. Given the crucial role played by autophagy in the RPE, alteration of the autophagy flux could have deleterious consequences for the cells, including accumulation of toxic compounds (i.e., vacuoles containing undigested materials, oxidized compounds, aggregates, and damaged organelles) and uncontrolled inflammation, which are conditions contributing to retinal degeneration. Data from the literature indicate that lutein and curcumin can modulate autophagy [12,38,72]. Our data suggest that lutein, and potentially curcumin can prevent autophagy flux alteration induced by blue LED exposure. We have to keep in mind that autophagy can have a dual role [73] Recently, it has been shown that autophagy-associated cell death was induced in ARPE-19 and primary human RPE (hRPE) cells by serum deprivation and oxidative stress by H_2_O_2_, and that the clearance of these cells induced inflammation, suggesting this mechanism could contribute to AMD [74]. Therefore, it would be interesting to evaluate the effects of curcumin and lutein in this mechanism. 

Most of the time, dietary supplements are composed of a complex mixture. Therefore, we wanted to investigate the interactions between molecules leading to antagonistic or synergistic effects. The combination of curcumin and lutein decreased metabolic activity (previously observed with curcumin) and limited oxidative stress induced by H_2_O_2_ and toxic effects of staurosporin, and prevented VEGF release induced by CoCl_2_. The association of curcumin (0.34 or 3.4 µM) and lutein (3.5 µM) with or without resveratrol did not show toxic effects, but no antagonistic or synergistic effect between agents could be found. Nevertheless, we did observe a complementary protective effect. 

## 5. Conclusions

In this study, we analyzed the ability of lutein, curcumin, and resveratrol to protect human retinal pigment epithelial cells (ARPE-19 cells) against several harmful stressors (oxidative stress, induced cell death, hypoxia, and autophagy alteration induced by blue LED) known to be involved in retinal degeneration. We could not find evidence of any protective effect of resveratrol (in the range of 0.001 to 21.9 µM). Curcumin had higher protective effects against H_2_O_2_-induced oxidative stress than lutein. At the concentration tested, lutein and curcumin had a similar protective effect against staurosporine-induced cell death. Pretreatment with lutein, but not curcumin was found to prevent blue LED-induced alteration of autophagy. In addition, whereas curcumin had no effect on basal VEGF release from human-derived retinal cells, lutein reduced it. Therefore, because of the complementary effect of each compound, the association of lutein and curcumin is beneficial to protect human retinal pigment epithelial cells. The combination of curcumin and lutein should reduce the effect of oxidative stress on metabolic activity, apoptotic cell death, and VEGF secretion induced by hypoxia. 

## Figures and Tables

**Figure 1 nutrients-12-00879-f001:**
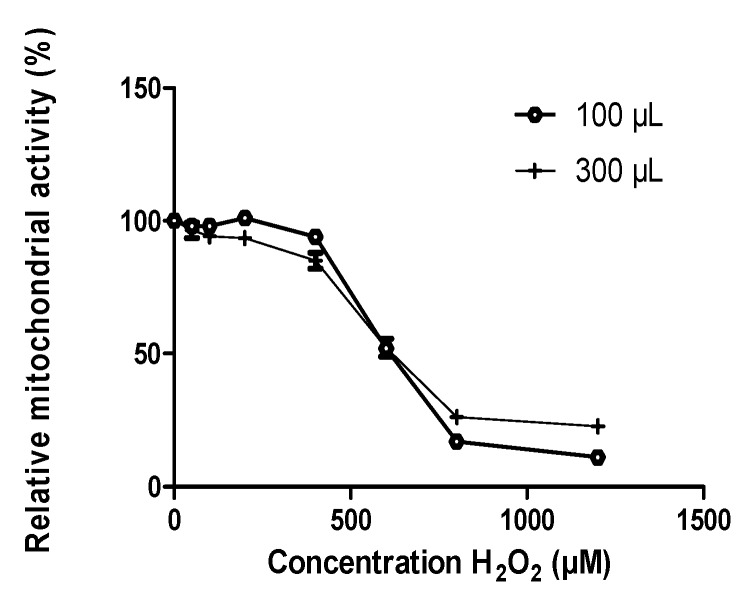
Effects of H_2_O_2_ on ARPE-19 cells. Plates were seeded with 100 or 300 µL per well of cell suspensions at 2.10^5^ cells/mL After 3 days, cells were treated with H_2_O_2_ for 2 h. Metabolic activity was evaluated by MTT assay. Results are presented as relative mean metabolic activity ± SEM.

**Figure 2 nutrients-12-00879-f002:**
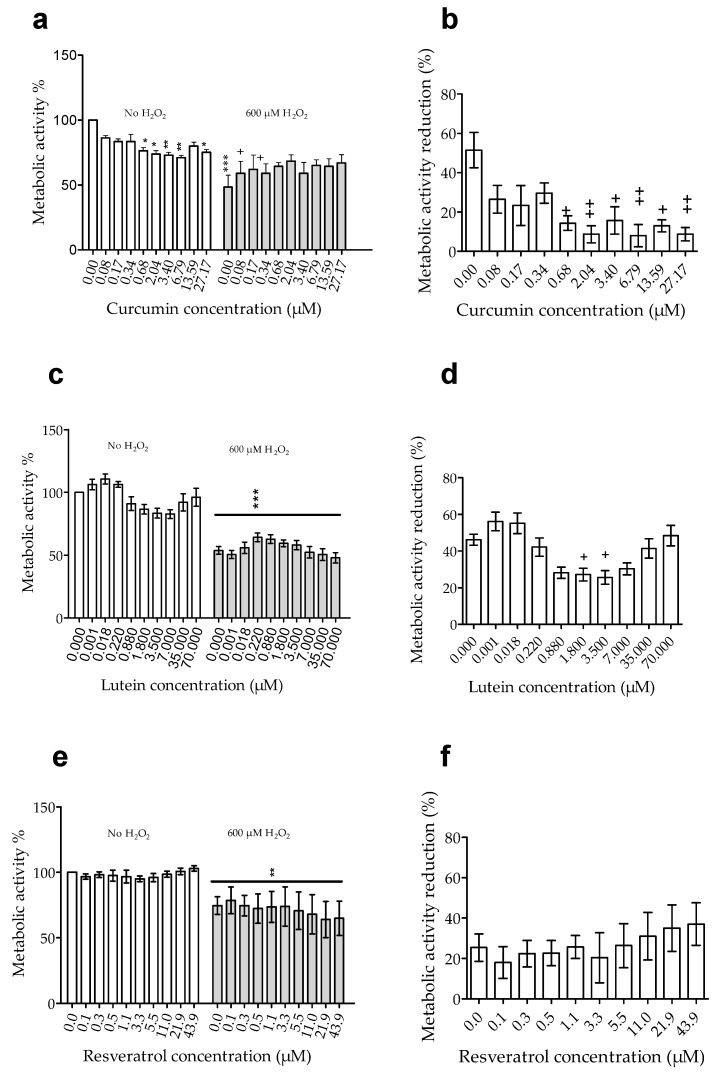
Effect of curcumin, lutein, and resveratrol on oxidative stress induced by H_2_O_2_. ARPE-19 cells were untreated (0) or pre-treated with different concentrations of curcumin (**a,b**), lutein (**c,d**), or resveratrol (**e,f**) before adding 600 µM H_2_O_2_ or No-H_2_O_2_. Metabolic activity was quantified by MTT assay. Left **(a,c,e):** Relative metabolic activity with No treatment (0.0)-No H_2_O_2_ (left side of the graph) being 100%. Right **(b,d,f):** Metabolic activity reduction corresponding to metabolic activity of cells with No-H_2_O_2_ minus the one of cells with 600 µM H_2_O_2_ for each treatment. Results are expressed as mean ± SEM (*n* = 4 to 11 for lutein; 3 to 5 for curcumin and 3-4 for resveratrol, where each test was performed in triplicates). (*) compared to no treatment-No H_2_O_2,_ (+) compared to No treatment-600 µM H_2_O_2._

**Figure 3 nutrients-12-00879-f003:**
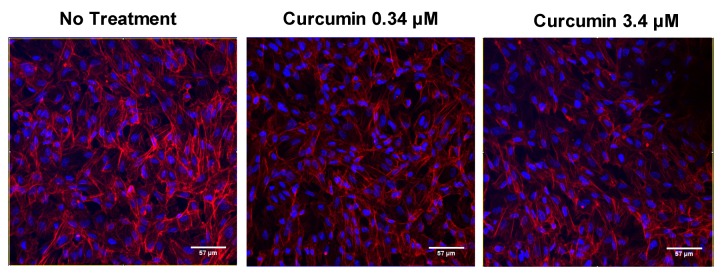
Effect of curcumin on the morphology of ARPE-19 cells. Representative images of DAPI/ Phalloidin labelling in untreated and treated cells with curcumin at 0.34 or 3.4 µM.

**Figure 4 nutrients-12-00879-f004:**
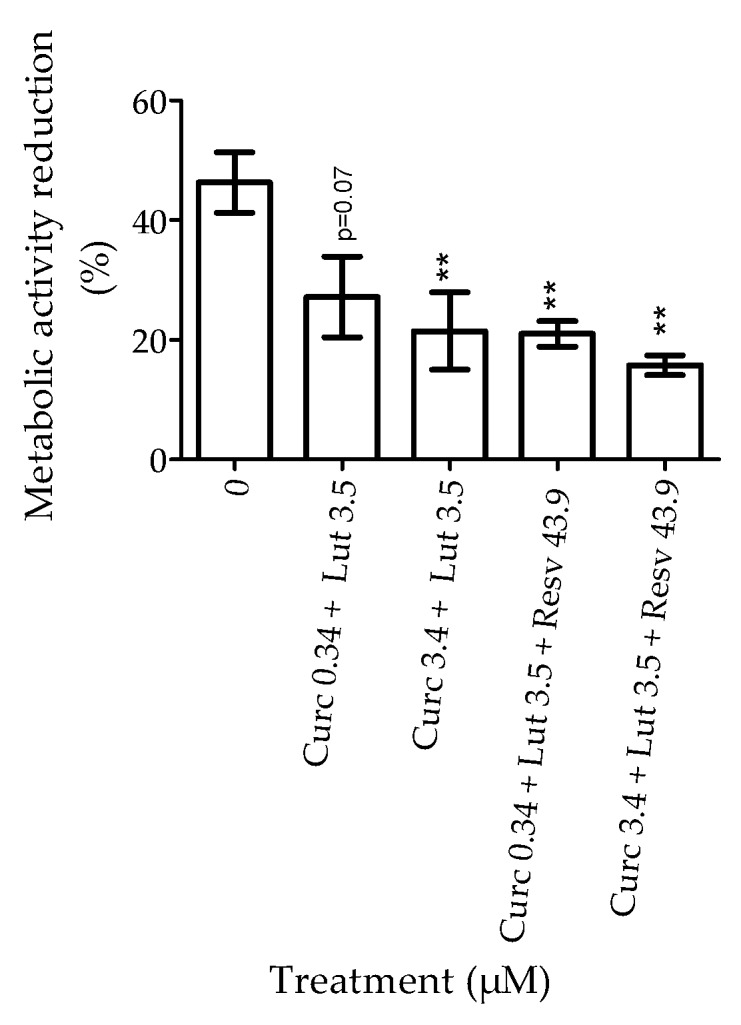
Effect of pretreatment with combination of natural agents on the reduction of metabolic activity induced by H_2_O_2_. Results are expressed as percentage ± SEM; *n* = 7 (each experiment was performed in triplicate). Statistical significance (*) compared to no pretreatment.

**Figure 5 nutrients-12-00879-f005:**
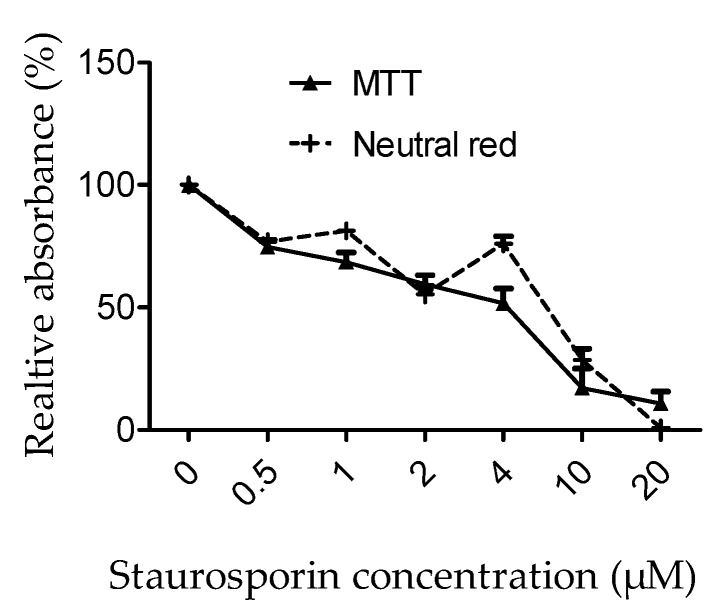
Staurosporin-induced apoptosis in ARPE-19 cells. Cells were treated with different concentrations of staurosporin before evaluating metabolic activity (MTT) and cell viability (neutral red). Results are expressed as mean % ± SEM (*n* = 4–5 independent experiments).

**Figure 6 nutrients-12-00879-f006:**
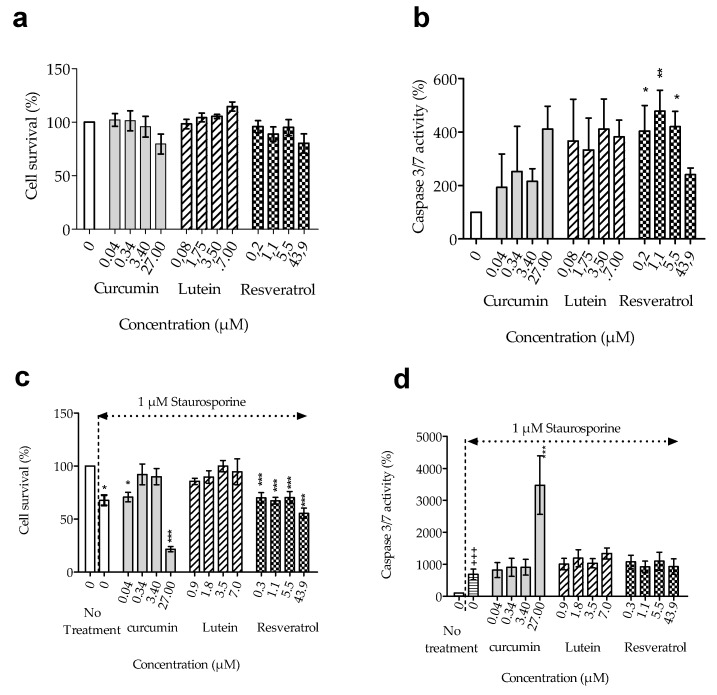
Effect of curcumin, lutein, and resveratrol on cell survival and caspase activity. (**a**) Cell survival and (**b**) caspase activity in ARPE-19 cells pre-treated for 24 h with different concentrations of curcumin, lutein, or resveratrol. (**c**) Cell survival and (**d**) caspase activity in cells pretreated for 24 hours with different concentrations of curcumin, lutein, or resveratrol and then exposed to staurosporin at 1 µM for 18 h. Results are expressed as mean % ± SEM (*n* = 4–5 independent experiments). Statistical significance (*) compared to no pretreatment.

**Figure 7 nutrients-12-00879-f007:**
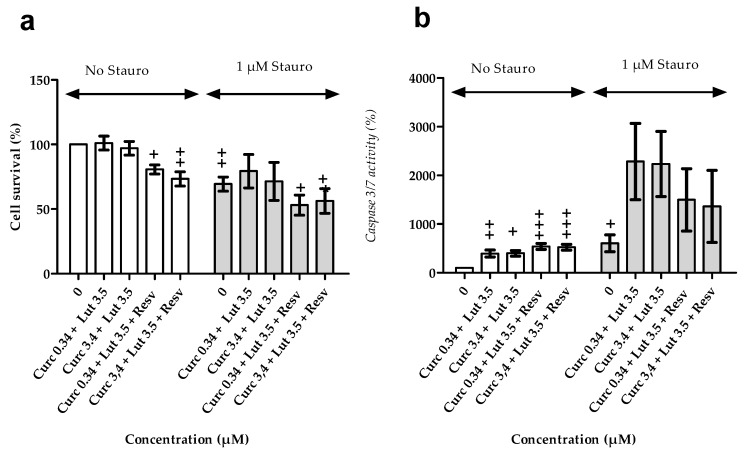
Effect of combination of natural agent on cell survival and caspase 3/7 activity. ARPE-19 cells were treated with curcumin + lutein or with curcumin + lutein + resveratrol for 24 h. Then, they were treated with staurosporin at 1 µM for 18 h. (**a**) Cell survival assessment and (**b**) caspase 3/7 activity. Results are expressed as mean percentage ± SEM (n = 6 independent experiments, each tested in triplicate). Statistical significance (+) compared to no pre-treatment.

**Figure 8 nutrients-12-00879-f008:**
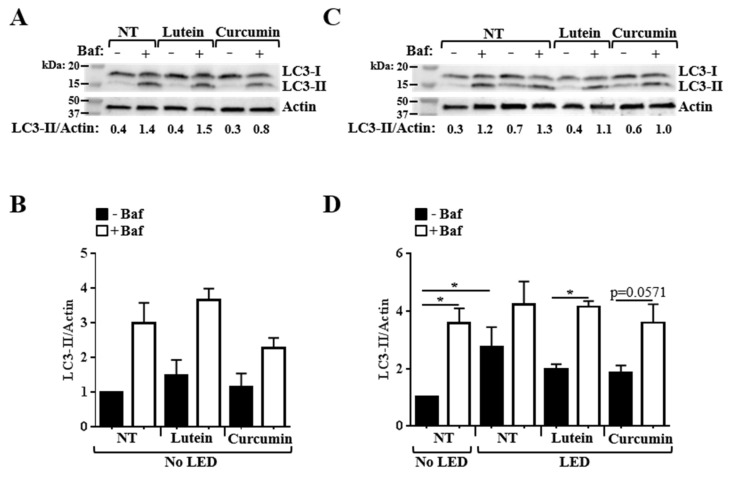
Effect of lutein and curcumin on autophagy. ARPE-19 cells were treated with lutein at 3.5 µM and curcumin at 3.4 µM for 24 h before being exposed or not to blue LED for 4 h. Bafilomycin A1 (Baf) was added 2 h before protein extraction. Immunoblot analyses were performed using anti-LC3B and anti-Actin antibodies. Quantification of LC3-II and Actin was done. (**A**,**C**) are representative immunoblots. (**B**,**D**) are the corresponding ratios of LC3-II/Actin normalized to that obtained for cells that were untreated, non-exposed to blue LED and without Baf, defined as 1.0. Data are means ± SEM of at least 3–4 independent experiments. For statistical analysis, comparisons were performed between untreated cells with Baf and cells treated with lutein or curcumin, with Baf by using the non-parametric Mann–Whitney test. Comparisons were also performed between cells without Baf or cells with Baf, in the different conditions (untreated, or treated with lutein or curcumin), by using the non-parametric Kruskal–Wallis test. * *p* < 0.05.

**Figure 9 nutrients-12-00879-f009:**
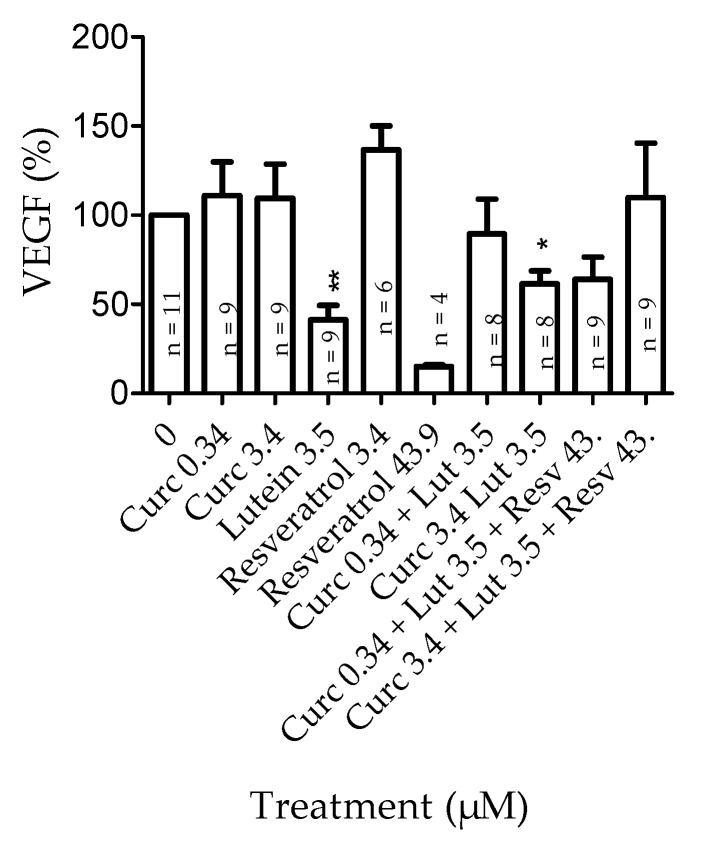
Effect of curcumin, lutein, and resveratrol on vascular endothelial growth factor (VEGF) secretion. VEGF amounts were quantified by ELISA in the cell supernatants after a 24 h incubation with natural agents, alone or combined. Results are expressed as mean percentage ± SEM. The number of tests for each condition is indicated in each bar. Each *n* corresponds to a triplicate. Statistical significance (*) compared to “0”.

**Figure 10 nutrients-12-00879-f010:**
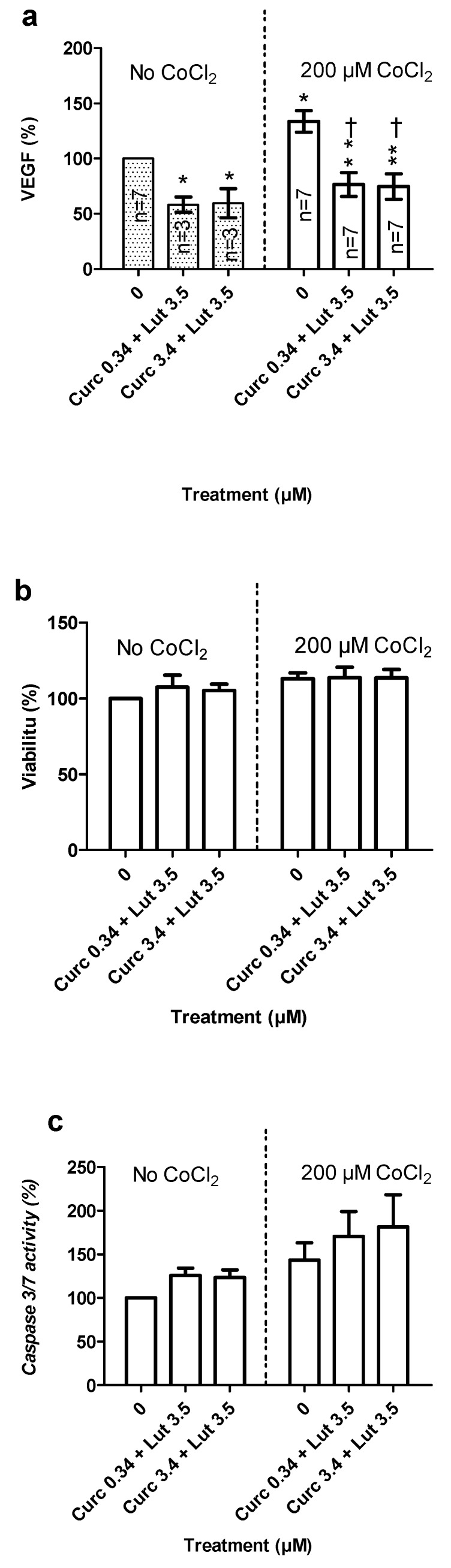
Effect of curcumin and lutein on the prevention of hypoxia-induced VEGF secretion. ARPE-19 cells were treated or untreated with curcumin + lutein for 24 h before induction of hypoxia with CoCl_2_ at 200 µM for 4 h. (**a**). VEGF quantification by ELISA in supernatants. (**b**). Cell viability and (**c**). caspase 3/7 activity (*n* = 4, triplicates per group). Results are expressed as mean % ± SEM. Statistical significance (*) compared to No-CoCl_2_-No-pretreatment or (†) compared to CoCl_2_-No-Pretreatment.

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
