# Peer review of "Cytoprotective Effects of Natural Highly Bio-Available Vegetable Derivatives on Human-Derived Retinal Cells"

_nutrients, 2020, doi:10.3390/nu12030879_

Round 1

Reviewer 1 Report

The present paper need a reorganization of the experimental data, to be presented in a more simplified mode.

Figure 2 a, c, e need to be edited; it is hard to be understood.

Figure 3 need to be adjusted the space between the 3 picture and the corresponding dose.

Figure 4 needs to be presented in a simplified mode; it cannot understand the different combination. Any synergic effects?

Can be calculated the synergism between the phytochemical combination? Any dose-effect response can be observed?

Novel mechanistic insight presented in this paper? Can be summarized as a new figure?

Additional discussion related to synergic/antagonic effect of the tested bioactive agents.

Author Response

file

Reviewer 2 Report

Munia et al reported the results of in vitro experiments using ARPE19 cell line. They analyzed the effects of curcumin, lutein, resveratrol, and their combination on the ARPE19 cells with or without stimuli of H2O2 or staurosporin or blue LED light or CoCL2. The results were only listed and the relationships between the data were unclear. Also the demonstration of the data, in particular, Figure 2 was complicated.

It was hard to understand how they drew the conclusion that at least curcumin and lutein were beneficial, although in most experiments, curcumin, and lutein seemed to be toxic under no stimuli.

Figure 2b, d, f may better be shown by % metabolic activity, rather than metabolic activity reduction. Were the data compared between the values of the experiments with and without H2O2 in each experiment of adding curcumin or lutein or resveratrol?

There were no mechanisms in either experiments. For example, how were the oxidative stress levels?

They described about necrosis in Figure 3 in which phalloidin and DAPI were used for staining. How did they judge whether there was necrosis?

It is hard to understand that they solubilized lutein in water. It is not water soluble.

The goal of the current study was unclear. What did they want to reveal by listing several experiments?

Author Response

Dear Reviewer,

please find the answers as an attached file.

Best regards,

The authors

Reviewer 3 Report

This is a sound and well performed study.

Major Points:

1. The authors should at least argue on the applied lutein, curcumin and resveratrol concentrations. In principle, a measure of the intracellular effective concentrations of Lutein, curcumin and resveratrol would have been needed when comparing the efficacy of These compounds.

2. The authors should explain / argue why they selected neutral red as viability marker.

minor: line 95 dose (µM is missing)

Author Response

(The authors gave the same response as above.)

Reviewer 4 Report

This manuscript contains extensive amount of in vitro data which support cytoprotective effects of natural compounds on a retinal pigment epithelial cell line, ARPE-19. The study is clinically relevant and would be even more valuable if the most important results were confirmed with primary cells.

Comments:

  1. There are grammatical errors in lanes 229, 288, 390, 441, 443 and 451.
  2. In lanes 204 and 205, the text cites Figure 4 instead of Figure 3.
  3. The logic of the text sometimes does not follow the order of the (sub)figures (lanes 202-209 and 264-282).
  4. Figures 5 and 6; 9 and 10 might be combined.
  5. The measured parameter in Figures 1, 2 and 4 is the same. However, in Figure 1 it is labelled as “Relative mitochondrial activity (%)” and in Figures 2 and 4 as “Metabolic activity %”; this should be unified.
  6. In the legends of Figures 2, 6 and 7, the meanings of * and + symbols should be provided. Above the last bar of Figure 7a, the labelling of significance is not clearly visible.
  7. In Figures 3 and 4c, images with larger magnifications and scale bars should be provided.
  8. In Figure 8, authors should provide which isoform of the Actin was immunoblotted. Furthermore, size markers should be also included.
  9. It was shown by Szatmári-Tóth M et al. in primary and cell line models of retinal pigment epithelial cells (Cell Death Dis, 2016; Int J Mol Sci, 2019), that oxidative stress induced by hydrogen peroxide resulted in autophagy-associated cell death which can contribute to the pathogenesis of Age-Related Macular Degeneration. This should be discussed in the manuscript.

Author Response

(The authors gave the same response as above.)

Reviewer 5 Report

An excellent study on the effects of plant compounds that attenuate oxidative stress on RPE culture cells. Suggested next steps include nutritional and bioavailability studies to assess the presence or absence of these compounds for in vivo human RPE following ingestion. 

Author Response

(The authors gave the same response as above.)

Round 2

Reviewer 1 Report

Please keep only the data that are statistically significant in the paper. the data that are not statistically significant present as supplimentary fille.

Please make the disscussion only around the data that are statistically significant, emhasis the therapeutic combination, the contribution on the therapeutic effect.

Author Response

See the file attached

Reviewer 2 Report

The manuscript has been improved.
